# Influence of Critical Parameters on the Extraction of Concentrated C-PE from Thermotolerant Cyanobacteria

**DOI:** 10.3390/biotech13030021

**Published:** 2024-06-24

**Authors:** Ariadna H. Vergel-Suarez, Janet B. García-Martínez, German L. López-Barrera, Néstor A. Urbina-Suarez, Andrés F. Barajas-Solano

**Affiliations:** Department of Environmental Sciences, Universidad Francisco de Paula Santander, Av. Gran Colombia No. 12E-96, Cúcuta 540003, Colombia; ariadnahazelvs@ufps.edu.co (A.H.V.-S.); janetbibianagm@ufps.edu.co (J.B.G.-M.); lucianolb@ufps.edu.co (G.L.L.-B.); nestorandresus@ufps.edu.co (N.A.U.-S.)

**Keywords:** cyanobacteria, phycobiliproteins, natural colorants, biomass dehydration

## Abstract

This work aimed to identify the influence of pH, molarity, *w*/*v* fraction, extraction time, agitation, and either a sodium (Na_2_HPO_4_·7H_2_O-NaH_2_PO_4_·H_2_O) or potassium buffer (K_2_HPO_4_-KH_2_PO_4_) used in the extraction of C-phycoerythrin (C-PE) from a thermotolerant strain of *Potamosiphon* sp. An experimental design (Minimum Run Resolution V Factorial Design) and a Central Composite Design (CCD) were used. According to the statistical results of the first design, the K-PO_4_ buffer, pH, molarity, and *w*/*v* fraction are vital factors that enhance the extractability of C-PE. The construction of a CCD design of the experiments suggests that the potassium phosphate buffer at pH 5.8, longer extraction times (50 min), and minimal extraction speed (1000 rpm) are ideal for maximizing C-PE concentration, while purity is unaffected by the design conditions. This optimization improves extraction yields and maintains the desired bright purple color of the phycobiliprotein.

## 1. Introduction

Cyanobacteria are a group of photosynthetic microorganisms that belong to the bacterial kingdom, although they share some characteristics with algae. Like green plants, they are known for their ability to carry out photosynthesis and produce oxygen. However, unlike plants, cyanobacteria do not have chloroplasts and lack a defined nucleus, distinguishing them as bacteria [1,2,3]. In scientific research and biotechnology, cyanobacteria have also proven helpful because of their ability to produce a variety of valuable compounds, such as biofuels, pharmaceuticals, industrial chemicals, and biofertilizers, through genetic engineering and metabolic modifications [4,5,6,7,8,9].

Phycobiliproteins (PBPs) are abundant water-soluble proteins that cyanobacteria produce to capture the light energy that is transferred to the chlorophyll during photosynthesis [10,11]. PBPs are formed by a complex between covalently bound proteins and phycobilins, which trap light and are the significant components of phycobilisomes [12,13]. The following four categories can be distinguished between phycobiliproteins based on how well they absorb light: Phycoerythrins (490–570 nm), phycocyanins (610–625 nm), phycoerythrocyanins (560–600 nm), and allophycocyanins (650–660 nm) [14,15]. C-phycoerythrin (C-PE) is a water-soluble protein pigment complex found mainly in red algae and, together with phycocyanin and allophycocyanin, belongs to the photosynthetic pigments and forms an accessory light-collecting apparatus [16]. C-phycoerythrin (C-PE) has great potential for applications in the food, pharmaceutical, and cosmetic industries. PE has demonstrated several medical uses in addition to its employment as a synthetic dye substitute in the conventional food sector. For instance, C-PE’s fluorescent qualities make it worthwhile for cancer diagnosis and HIV monitoring. [17].

Among the main functions of C-phycoerythrins (C-PE) is their participation in nitrogen reserve processes; when cyanobacteria are faced with a nitrogen deficiency, they can degrade phycobiliproteins that contain a high concentration of nitrogen, such as C-phycoerythrin, and can also act as a quantum collector for photosynthesis. They absorb light at wavelengths inaccessible to chlorophyll and transfers the energy to the photosynthetic reaction center [18]. Some cyanobacteria can modify the chromophore composition of phycoerythrin by the complementary chromatic adaptation process, allowing them to acclimate to variations in light intensity and quality [19]. *Potamosiphon* sp. is a novel genus previously isolated in a marine environment in Australia [20]; however, in 2022, our research group was able to isolate a strain from a thermal spring. The strain can grow under high light intensities and high temperatures (30–45 °C) while producing a high concentration of C-PE [21].

The amount of C-phycoerythrin (C-PE) present in cyanobacterial cells varies within different species and culture environments. Cyanobacteria generally maintain cell development and activity in nitrogen deficiency by degrading phycobiliproteins that contain a high nitrogen concentration [22]. When some cyanobacterial strains are acclimated to specific light intensity and wavelength, the complementary chromatic adaptation process allows for the modification of the chromophore composition of C-PE [23,24].

Based on the physicochemical, structural, and spectroscopic properties of phycoerythrin, various biotechnological applications have been proposed, as well as new extraction methods that consider the solubility in water at room temperature and the sensitivity of the pigment to temperatures above 60 °C [25]. Problems associated with large polysaccharides (e.g., agar and cellulose) in the cell wall present a significant obstacle to cell disruption during the primary extraction of metabolites [26], partly because they form a complex matrix. This matrix adds strength and rigidity to the cell, reducing the extractability of biomolecules, and is also due to the strong covalent bond between the xylan bound to the mixture and the glycoprotein complex [27]. Therefore, a suitable cell disruption method is required for extraction. Several methods have been described to extract high-quality products from red cyanobacteria. Several conventional buffers have been tested to extract C-phycoerythrin (C-PE) [28]. The following three extraction solvents have been tested: sodium buffer (Na_2_HPO_4_·7H_2_O-NaH_2_PO_4_·H_2_O), CaCl_2_ solution, and distilled water. Mittal et al. [16] used a potassium buffer (K_2_HPO_4_-KH_2_PO_4_, 0.1 M, pH 6.8), and Li et al. [29] used a mixture of cold distilled water with 95% ethanol, followed by hot distilled water to extract PE, lipids, and polysaccharides. Finally, García et al. [30] used PO_4_ buffer (0.1 M, pH 5.5), followed by precipitation with ammonium sulfate ((NH_4_)_2_SO_4_) at 60% saturation.

Solvent extraction methods are insufficient to achieve optimal metabolite concentrations. Several authors have tried extraction methods using osmotic shock [31], maceration in the presence of liquid nitrogen [32], freeze–trituration [33], freeze–thaw [34], ultrasound [35], and homogenization [36]. These methods may become uneconomical for long extraction times. Therefore, a suitable cell disruption method for extraction should combine different buffers and mechanical processes, such as vortex stirring, and evaluating stirring times, pH, and molarity.

This study aimed to develop an efficient method for extracting C-phycoerythrin (C-PE) from a new strain of thermotolerant *Potamosiphon* sp. An experimental design was used to evaluate the influence of pH, molarity, *w*/*v* fraction, extraction time, and agitation on the type of buffer used to extract the highest amount of phycoerythrin (C-PE). We have previously reported the effect of the drying on the extraction efficiency of C-phycoerythrin (C-PE) [21].

## 2. Materials and Methods

### 2.1. Strain

*Potamosiphon* sp. UFPS003 was isolated from a thermal spring near Cucuta (Colombia). The strain was kept in solid BG-11 culture media [37] at the INNOValgae collection (UFPS, Cucuta, Colombia) (https://www.innovalg.com accessed on 5 June 2024). The strain was initially grown in 0.2 L of BG-11 culture media in a 0.5 L Schott GL45 glass flask. The media was aerated by injecting filtered air enriched with 1% (*v*/*v*) CO_2_ at a flow rate of 0.12 L min ^−1^. The culture was exposed to cool-white LED lamps at an intensity of 100 µmol m ^−2^ s ^−1^, 12:12 h photoperiod, and a temperature of 27 ± 1 °C for 15 days.

### 2.2. Experimental Design

The effect of the critical variables (Table 1) on the purity and concentration of the extracted C-PE was analyzed using a Minimum Run Resolution V Factorial Design with six variables (five numeric and one categoric), coupled with a surface response in Design-Expert^®^ software (version 13.0, Stat-Ease Inc., Minneapolis, MN, USA). The different variables and their levels are presented in Table 1, while Table A1 shows the resolved design with the 22 evaluated experiments. 

### 2.3. Culture Conditions

The strain was cultured in triplicate (original plus two replicates) for each experiment in 0.5 L Schott GL45 flasks with a working volume of 0.2 L of liquid BG-11 media. Each flask was connected to a compressed air line mixed with CO_2_ (1% *w*/*w*) at 0.12 L_air_ min^−1^, 12:12 h photoperiod, at 100 µmol m^−2^ s^−1^, and 27 ± 1 °C for 20 days.

### 2.4. Biomass Drying, C-PE Extraction, and Quantification

Each flask was disconnected from the compressed air and allowed to settle for about twenty minutes. The supernatant was axenically removed, and the precipitated biomass was further concentrated by centrifugation (3600 rpm, 10 °C, 20 min). The harvested biomass was dried according to Vergel-Suarez et al. [21], using a food-grade dehydrator (40 °C, 12 h). The dried biomass was weighted and used for the extraction experiments. A known amount of dried biomass was mixed with a known volume of cold buffer solution (sodium buffer, (Na_2_HPO_4_·7H_2_O-NaH_2_PO_4_·H_2_O) or potassium buffer (K_2_HPO_4_-KH_2_PO_4_)) until it reached the biomass/solvent ratio, as shown in Table 2. The sample of biomass and buffer was also mixed with a known amount of glass beads (0.5 mm diameter), following the method described by Barajas-Solano [38]. The mixture was mixed using an automatic vortex (Multi Reax, Heidolph, Germany) according to the conditions of each experiment. The mixture was precipitated in a refrigerator overnight. Finally, the deep-purple extract rich in C-PE was separated by centrifugation (3600 rpm, 20 min, 10 °C). 

The concentration of C-PE was calculated using Equations (1)–(3), previously described by Bennett and Bogorad [39], while its purity was obtained using Equation (4) [40,41]. The mean obtained for each experiment was used for the ANOVA analysis according to the Design-Expert^®^ software.
(1)$$C-\mathrm{PC}\left(\frac{g}{L}\right)=\frac{{OD}_{620nm}-0.474\left({OD}_{652nm}\right)}{5.34}$$
(2)$$\mathrm{APC}\left(\frac{g}{L}\right)=\frac{{OD}_{652nm}-0.208\left({OD}_{620nm}\right)}{5.09}$$
(3)$$C-\mathrm{PE}\left(\frac{g}{L}\right)=\frac{{(OD}_{562nm}-2.41\left(C-PC\right)-0.849(APC))}{9.62}$$
(4)$$C-\mathrm{PE}[\mathrm{purity}]=\frac{{OD}_{562nm}}{280}$$

### 2.5. Process Optimization

The most relevant variables to improve the concentration and purity of extracted C-PE were further investigated using a Central Composite Design (CCD). All experiments proposed in the optimization were performed six times (original plus five replicates).

## 3. Results

### 3.1. Effect of Multiple Variables on the Concentration and Purity of C-PE

The ANOVA analysis of the experimental data is highlighted in Table 2. According to the results, the variables that significantly affect (*p* < 0.05) the concentration of C-PE are biomass/buffer ratio (A), pH (B), molarity (C), and the interactions between two variables, such as AC, BE, DE, CE. However, extraction time and speed were not statistically relevant to the process. The model’s F-value of 21.13 implies that the obtained model is significant, with only a 0.01% chance that the F-value obtained could occur due to the noise of the different experiments. On the other hand, the low difference between the Predicted and the Adjusted R^2^ is less than 0.2, and the Adeq Precision of 17.485 indicates an adequate signal-to-noise ratio. Therefore, the data obtained for the effect of the multiple variables on the concentration of C-PE are enough to improve the content of the extracted protein.

According to the ANOVA for C-PE purity (Table 3), the variables that significantly affect (*p* < 0.05) the concentration and purity of C-PE are biomass/buffer ratio (A), pH (B), extraction time (D), extraction speed (E), and buffer used (F) and the interactions between two variables, such as AB, AC, BD, BE, CE, CF, DE. However, the molarity of the buffer was not statistically relevant to the process. The model’s F-value of 78.92 implies that the obtained model is significant, with only a 0.01% chance that the F-value obtained could occur due to the noise of the different experiments. On the other hand, the low difference between the Predicted and the Adjusted R^2^ is less than 0.2, and the Adeq Precision of 40.011 indicates an adequate signal-to-noise ratio. Therefore, the data obtained for the effect of the multiple variables on the concentration of C-PE are enough to improve the purity.

### 3.2. Optimization of Relevant Variables

According to the data obtained, the best possible scenario to enhance the concentration and purity of C-PE requires a lower biomass/buffer ratio, a pH closer to 6.3, a molarity of 0.001, longer extraction times, higher extraction speed, and a potassium phosphate (K_2_HPO_4_-KH_2_PO_4_) buffer. Based on the obtained results, a new design, specifically a Central Composite Design of three variables (pH, extraction speed, and time), was used to optimize the concentration and purity of the extracted C-PE (Table 4). The resolved design with the experimental data can be found in Table A2.

The results for the concentration and purity of the extracted C-PE from the optimization design are presented in Table 5. According to the results obtained for the concentration of C-PE, the model’s F-value (339.73) is significant, and the noise does not affect the data. The most significant terms (*p* < 0.05) were the extraction time and speed, their quadratic counterparts (A^2^, B^2^, C^2^), and several of the interactions (AB, AC, BC). The R^2^ obtained shows a good fit (0.9974). Also, the difference between the adjusted R^2^ (0.9945) and the predicted R^2^ (0.9830) was less than 0.2. In the case of the data obtained for the purity of the extracted C-PE, the model’s F-value is also significant (53.46). Unlike the concentration, the pH of the buffer only affects the purity. Finally, the R^2^ obtained shows a good fit (0.9197). Also, the difference between the adjusted R^2^ (0.9025) and the predicted R^2^ (0.8389) is less than 0.2.

Figure 1 shows the response surfaces for the concentration (Figure 1a) and purity (Figure 1b) of C-PE. In the case of the concentration of C-PE, longer times and lower pH significantly improve the final content of C-PE; however, on the purity of the extract, there is no curvature, and only the pH increases the result, while the time does not affect the outcome. It should be noted that the purity index obtained is higher than 0.7, which is considered food-grade. 

Using the optimization feature of Design-Expert^®^ software, the best scenario that maximizes C-PE concentration and purity is presented in Table 6. Those variables were tested using fresh biomass (under the same conditions as the other experiments).

The results were analyzed through a one-sample *t*-test using GraphPad Prism version 10.2.3 for Mac (GraphPad Software, www.graphpad.com (accessed on 10 June 2024)) (Figure 2a,b). The analysis shows that the proposed method improves both the concentration and purity of the extracted C-PE while retaining a brilliant purple color (Figure 2c).

## 4. Discussion

Due to their thermolability, extracting any phycobiliprotein is a critical step that requires fine-tuning according to the specific requirements of the target protein. Multiple variables such as extraction time, speed pH, and the selected buffer are the most used to improve the extraction of phycobiliproteins [42,43]. In the case of extraction time and speed, it is always preferable to obtain the conditions that transmit the least energy to the sample, which can generate significant losses due to the rapid degradation of specific proteins at temperatures above 60 °C. According to the literature, the most common extraction times range between 15 and 20 min [43,44,45], while the reported speed used was from 2000 to 10,000 rpm. Now, most of those conditions have been adjusted to multiple cycles of freeze–thaw of the biomass, such as those reported by Ji et al. [46] However, in this case, the biomass was first dehydrated to reduce the cellular moisture as much as possible while improving the extracted content of C-PE [21]. 

Over the years, several buffers have been tested as viable solvents for extracting all phycobiliproteins. Buffers such as sodium phosphate (NaHPO_4_-Na_2_HPO_4_), potassium phosphate (K_2_HPO_4_-KH_2_PO_4_), Tris-HCl, and even CaCl_2_ have been tested [47,48]. However, according to Pez Jaeschke et al. [47], the preferred extraction buffer should be acidic (between 5 and 6); in this case, the results show that an acidic potassium phosphate buffer (pH 5.8) allows for a good extraction of C-PE from the biomass. Other variables used in the initial design showed no statistical significance in concentration and purity extraction.

An interesting result is that the buffer type only affects the purity rather than the concentration of C-PE extracted. The latter is relevant since most of the literature shows that the potassium phosphate buffer is preferred; however, to the author’s knowledge, no study has proven such a specific difference between the sodium phosphate (NaHPO_4_-Na_2_HPO_4_) and potassium phosphate (K_2_HPO_4_-KH_2_PO_4_) buffers.

In the case of the first design, it was found that the type of buffer used only affects the purity index and does not affect the concentration. Although in the literature, it is possible to see the use of different buffers, including sodium phosphate, potassium phosphate, Tris-HCl, and even CaCl_2_, the potassium phosphate buffer is perhaps the most used for both phycocyanins and phycoerythrins; however, as far as the authors know, no report explains why the potassium phosphate buffer is better than the others mentioned above. This is a significant result because it allows us to identify that in the case of C-PE, the potassium phosphate buffer, in comparison with the sodium phosphate buffer, selectively avoids secondary proteins, accessory or globular proteins, which maximizes both the concentration of C-PE and the purity of the crude extract.

In the case of C-PE concentration, the results indicate that the most important factors are the biomass–solvent ratio, pH, and molarity. Those factors that have almost no effect are extraction and time. In contrast, all the factors studied significantly influenced the purity index. In the specific case of this study, the biomass–solvent ratio and molarity factors were not explored in the optimization because the design aimed to use low concentrations of these two variables. A low biomass–solvent ratio will dilute the concentration of C-PE, significantly reducing its final content; similarly, a very low molarity (such as 0.001 mM) is close to zero, so it is not technologically relevant. This is why speed, extraction time, and pH were chosen to be optimized, as they are factors that can be increased and have a more significant effect. Optimization was performed in this case, and the results show that obtaining a considerable concentration and purity is possible. Similar conditions were obtained using *Lyngbya* sp. CCNM 2053 [49]; however, the final content of C-PE was higher in *Potamosiphon* sp. than in *Lyngbya* sp. CCNM 2053 (5% *w*/*w*) and *Microcoleus autumnalis* PACC 5522 [50].

Although statistical tools were used, such as in the design of experiments, the response surface for purity shows that, unlike the C-PE concentration, its purity is far from presenting results that can be considered optimal. It should be noted that the test results show that adjusting the extraction conditions makes it possible to obtain an extract with a considerable concentration of extracted C-PE and a purity of higher than 0.7 (considered as food-grade) [51,52]. At the same time, it is assumed that the conditions are to bring out as much protein as possible. In the case of purity, forcing a higher efficiency, i.e., a possible optimization at a more extended time and higher speed, is very complicated. All the conditions are given for as many of the water-soluble metabolites as possible, such as carbohydrates, glycoconjugates, accessory proteins, and other phycobiliproteins present in the phycobilisome such as C-PC and APC, to come out. So, we found that there is no possible optimization in purity, but there is a possible optimization in concentration.

## 5. Conclusions

The optimized extraction conditions suggest that a potassium phosphate buffer at pH 5.8, longer extraction times, and minimal extraction speed are ideal for maximizing C-PE concentration. At the same time, purity is unaffected by the design conditions studied. This optimization improves extraction yields and maintains the desired bright-purple color of the phycobiliprotein.

## Figures and Tables

**Figure 1 biotech-13-00021-f001:**
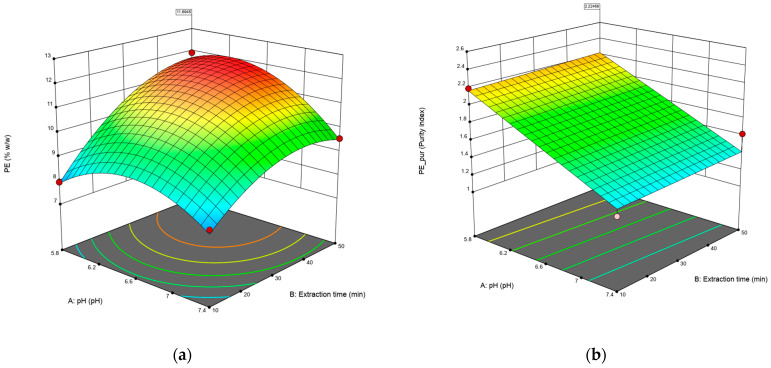
The surface response of the model was fitted to the data on concentration (% *w*/*w*) (**a**) and purity of C-PE (**b**). Red dots in figures represents design points used to create the surface.

**Figure 2 biotech-13-00021-f002:**
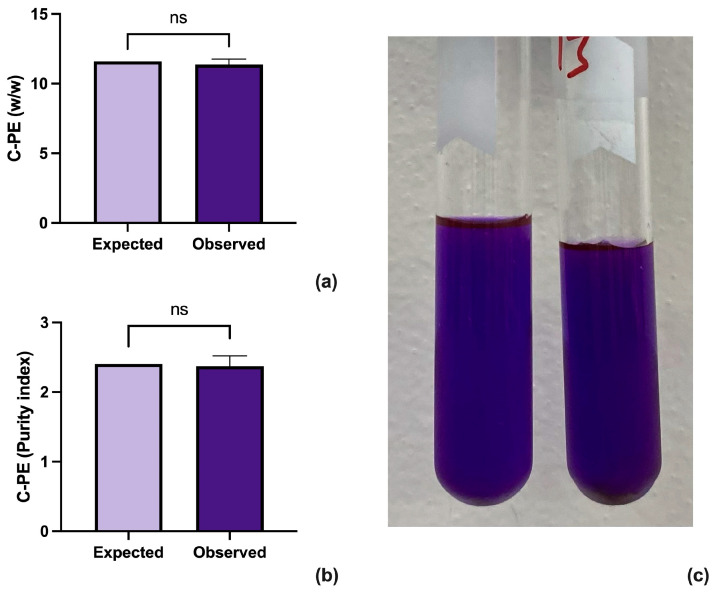
Results from the one sample *t*-test between the expected and observed results for concentration (**a**) and purity (**b**) of C-PE. A sample of highly concentrated sample of C-PE (**c**).

**Table 1 biotech-13-00021-t001:** Variables and their levels for the extraction of C-PE.

Coded Name	Variables	Units	Type	Low Level (−1)	High Level (+1)
A	Biomass/buffer ratio	% *w*/*v*	Numeric	1	10
B	pH	pH		5.8	7.40
C	Molarity	mM		0.0010	0.10
D	Extraction time	min		10	30
E	Extraction speed	rpm		500	1500
F	Buffer		Categoric	Na_2_HPO_4_·7H_2_O-NaH_2_PO_4_·H_2_O	K_2_HPO_4_-KH_2_PO_4_

**Table 2 biotech-13-00021-t002:** Analysis of variance (ANOVA) of the model obtained for C-PE concentration.

Source	Sum of Squares		Df	Mean Square		F-Value	*p*-Value
Model	6.42		9	0.7134		21.13	<0.0001 *
A-Biomass/buffer ratio	1.46		1	1.46		43.25	<0.0001 *
B-pH	0.1989		1	0.1989		5.89	0.0319 *
C-Molarity	0.1729		1	0.1729		5.12	0.0430 *
D-Extraction time	0.0507		1	0.0507		1.50	0.2442 **
E-Extraction speed	0.0391		1	0.0391		1.16	0.3028 **
AC	1.61		1	1.61		47.69	<0.0001 *
BE	1.34		1	1.34		39.74	<0.0001 *
CE	0.5514		1	0.5514		16.33	0.0016 *
DE	0.5977		1	0.5977		17.70	0.0012 *
Residual	0.4052		12	0.0338			
Cor Total	6.83		21				
	**R^2^**	**Adj R^2^**	**Pred R^2^**	**Adq Pr**	**Std. Dev.**	**Mean**	**C.V. %**
	0.9406	0.8961	0.7798	17.4849	0.1838	0.8436	21.78

* Significant. ** Not Significant.

**Table 3 biotech-13-00021-t003:** Analysis of variance (ANOVA) of the model obtained for C-PE purity.

Source	Sum of Squares		Df	Mean Square		F-Value	*p*-Value
Model	5.20		13	0.4003		78.92	<0.0001 *
A-Biomass/buffer ratio	0.0593		1	0.0593		11.68	0.0091 *
B-pH	1.25		1	1.25		247.07	<0.0001 *
C-Molarity	0.0012		1	0.0012		0.2275	0.6461 **
D-Extraction time	1.09		1	1.09		214.75	<0.0001 *
E-Extraction speed	0.5725		1	0.5725		112.85	<0.0001 *
F-Buffer	0.1301		1	0.1301		25.65	0.0010 *
AB	0.0953		1	0.0953		18.78	0.0025 *
AC	1.75		1	1.75		344.27	<0.0001 *
BD	0.3451		1	0.3451		68.03	<0.0001 *
BE	0.3432		1	0.3432		67.66	<0.0001 *
CE	0.0747		1	0.0747		14.72	0.0050 *
CF	0.1870		1	0.1870		36.86	0.0003 *
DE	1.47		1	1.47		289.59	<0.0001 *
Residual	0.0406		8	0.0051			
Cor Total	5.24		21				
	**R^2^**	**Adj R^2^**	**Pred R^2^**	**Adq Pr**	**Std. Dev.**	**Mean**	**C.V. %**
	0.9923	0.9797	0.9327	40.0111	0.0712	1.07	6.68

* Significant. ** Not Significant.

**Table 4 biotech-13-00021-t004:** Variables were evaluated based on their levels for the extraction of C-PE.

Coded Name	Variables	Units	Low Level (−1)	Center Point (0)	High Level (+1)
A	pH	pH	5.25	6.6	7.95
B	Extraction time	min	−3.64	30	63.64
C	Extraction speed	rpm	659.10	1500	2340

**Table 5 biotech-13-00021-t005:** Analysis of variance (ANOVA) of the optimization model for concentration and purity of extracted C-PE.

Response	Source	Sum of Squares		Df	Mean Square		F-Value	*p*-Value
C-PE (% *w*/*w*)	Block	0.4230		2	0.2115			
	Model	45.35		9	5.04		339.73	<0.0001 *
	A-pH	0.0405		1	0.0405		2.73	0.1372 **
	B-Extraction time	2.18		1	2.18		146.75	<0.0001 *
	C-Extraction speed	0.3108		1	0.3108		20.95	0.0018 *
	AB	2.32		1	2.32		156.21	<0.0001 *
	AC	2.01		1	2.01		135.19	<0.0001 *
	BC	7.48		1	7.48		504.14	<0.0001 *
	A^2^	20.25		1	20.25		1365.26	<0.0001 *
	B^2^	12.01		1	12.01		809.86	<0.0001 *
	C^2^	0.2743		1	0.2743		18.49	0.0026 *
	Residual	0.1187		8	0.0148			
	Lack of Fit	0.1064		5	0.0213		5.22	0.1021 **
	Pure Error	0.0122		3	0.0041			
	Cor Total	45.89		19				
		**R^2^**	**Adj R^2^**	**Pred R^2^**	**Adq Pr**	**Std. Dev.**	**Mean**	**C.V. %**
		0.9974	0.9945	0.9830	46.1151	0.1218	9.69	1.26
C-PE (% *w*/*w*)	Block	0.0811		2	0.0406			
	Model	2.32		3	0.7730		53.46	<0.0001 *
	A-pH	2.26		1	2.26		156.32	<0.0001 *
	B-Extraction time	0.0112		1	0.0112		0.7738	0.3939 **
	C-Extraction speed	0.0477		1	0.0477		3.30	0.0908 *
	Residual	0.2024		14	0.0145			
	Lack of Fit	0.1672		11	0.0152		1.29	0.4668 **
	Pure Error	0.0352		3	0.0117			
	Cor Total	2.60		19				
		**R^2^**	**Adj R^2^**	**Pred R^2^**	**Adq Pr**	**Std. Dev.**	**Mean**	**C.V. %**
		0.9197	0.9025	0.8389	20.7767	0.1202	1.73	6.96

* Significant. ** Not Significant.

**Table 6 biotech-13-00021-t006:** Best conditions to improve concentration and purity of extracted C-PE.

Coded Name	Variables	Units	Value
A	pH	pH	5.8
B	Extraction time	min	50
C	Extraction speed	rpm	1000
Z_1_	C-PE	*w*/*w*	11.6
Z_2_		Purity Index	2.4

## Data Availability

The original contributions presented in the study are included in the article.

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
