# Peer review of "Influence of Critical Parameters on the Extraction of Concentrated C-PE from Thermotolerant Cyanobacteria"

_biotech, 2024, doi:10.3390/biotech13030021_

Round 1

Reviewer 1 Report

Comments and Suggestions for Authors

Nice work! Good report. Can be useful for interested persons. I hope that you will find one or several applications using your method. Recommended for acceptance in present form.

Author Response

Assigned Editor

BIOTECH MDPI

June, 2024

Dear Editor

I would like to present the modified version of the paper entitled “Influence of critical parameters on the extraction of concentrated C-PE from thermotolerant cyanobacteria” according to the different comments suggested by reviewers.

You can find in the second version (V1) of the summited paper all the modifications highlighted in yellow the repeated sections modified (reviewer 1), green (reviewer 1), cyan (reviewer 2), and purple (reviewer 3).

Sincerely,

Prof. Andrés F. BARAJAS-SOLANO, PhD

Dep, Environmental Sciences

Universidad Francisco de Paula Santander

Cúcuta, Colombia.

Repeated sections.

The repeated sections pointed by the identity software are now highlighted in yellow

Reviewer 1 (Green).

Nice work! Good report. Can be useful for interested persons. I hope that you will find one or several applications using your method. Recommended for acceptance in present form.

Response.

We appreciate the kind response of reviewer 1.

Reviewer 2 (cyan).

This study aimed to develop an efficient method for extracting C- phycoerythrin (C- PE) from the newly isolated thermotolerant filamenous Potamosiphon sp. An experimental design was used to evaluate the influence of pH, Molarity, w/v fraction, extraction time, and agitation on the type of buffer used (Na-PO4 or K-PO4) to extract the highest amount of phycoerythrin (C- PE) using a Minimum Run Resolution V Factorial Design followed by a Central Composite Design (CCD). According to the statistical results of the first design, K-PO4 buffer, pH, Molarity, and w/v fraction are vital factors that enhance the extractability of C-PE. The construction of a CCD design of experiments suggests that K-PO4 buffer at pH 5.8, longer extraction times (50 min), and minimal extraction speed (1000 rpm) are ideal for maximizing C-PE concentration, while purity is unaffected by design conditions. This optimization improves extraction yields and maintains the desired bright purple color of the phycobiliprotein. The topic is novel but the application proposed is not so novel. The points needed addressed as follows:

(1) It is noted that your manuscript needs careful editing by someone with expertise in technical English editing paying particular attention to English grammar, spelling, and sentence structure so that the goals and results of the study are clear to the reader.

(2) The topic is novel but the application proposed is not so novel.

(3) It is advised to add the better extraction process under lower temperatures in process optimization.

Response:

(1). The document was revised using two AI (Grammarly and Curie), followed by a revision by an independent English native speaker.

(2). Even as the application so far is not novel, it is necessary to identify if this type of protein can be used for low or high purposes. According to literature, the main problem with PBP’s is the excess of impurities that can reduce the quality of the colorants.

(3). The method was done using cold buffer on a cold chamber (10 degrees). The data obtained showed no statistical difference under room temperature and cold chamber.

Reviewer 3 (purple).

This article presents a factorial design to improve critical parameters to improve the concentration of C-PE and purity from the recent isolated thermotolerant filamenous Potamosiphon sp.

The document is well structured and the methodologies are well implemented.

However there some points that I recommend the authors adress, in order to improve and enrich their paper.

  1. In introduction, I suggest to the authors to explain why they choose this novel cyanobacteria to make the optimization of C-PE extraction. Is this a rich-PE cyanobacteria? Is this described in literature? I recommend tou to explain, even briefly why you choose this specie.
  2. Regarding results, you have great focus on statistics, and that is ok. However, in my opinion there is a lack of presenting some results regarding the first factorial design. I would recommend you to present a table with conditions (runs) performed and the results regarding C-PE concentrations and purity index. For me and I guess for the readers too, would be easier to visualize the discrepancy/differences in results, than looking for the statistics itself.
  3. In any time you compare your the C-PE concentrations obained in optimized conditions of extraction with literature. Is high, low? Have potential to be apply in any biotechnological field? To enrich your discusion, I sugest you to compare the concentrations obtained  with values  in literature...I felt you never refer to this...

Response.

  1. A proper explanation has been added to the introduction section.
  2. The resolved designs (mini-run and CCD) with the data obtained are presented in appendix A.
  3. Data from other strains were added in discussion

Reviewer 2 Report

Comments and Suggestions for Authors

This study aimed to develop an efficient method for extracting C- phycoerythrin (C- PE) 12 from the newly isolated thermotolerant filamenous Potamosiphon sp. An experimental design was 13 used to evaluate the influence of pH, Molarity, w/v fraction, extraction time, and agitation on the 14 type of buffer used (Na-PO4 or K-PO4) to extract the highest amount of phycoerythrin (C- PE) using 15 a Minimum Run Resolution V Factorial Design followed by a Central Composite Design (CCD). 16 According to the statistical results of the first design, K-PO4 buffer, pH, Molarity, and w/v fraction 17 are vital factors that enhance the extractability of C-PE. The construction of a CCD design of exper- 18 iments suggests that K-PO4 buffer at pH 5.8, longer extraction times (50 min), and minimal extrac- 19 tion speed (1000 rpm) are ideal for maximizing C-PE concentration, while purity is unaffected by 20 design conditions. This optimization improves extraction yields and maintains the desired bright 21 purple color of the phycobiliprotein. The topic is novel but the application proposed is not so novel.The points needed addressed as follows:

(1)It is noted that your manuscript needs careful editing by someone with expertise in technical English editing paying particular attention to English grammar, spelling, and sentence structure so that the goals and results of the study are clear to the reader.

(2)The topic is novel but the application proposed is not so novel.

(3)It is advised to add the better extraction process under lower temperatures in process optimization.

Comments on the Quality of English Language

The quality of English needs more improved.

Author Response

(The authors gave the same response as above.)

Reviewer 3 Report

Comments and Suggestions for Authors

This article presents a factorial design to improve critical parameters to improve the concentration of C-PE and purity from the recent isolated thermotolerant filamenous Potamosiphon sp.

The document is well structured and the methodologies are well implemented.

However there some points that I recommend the authors adress, in order to improve and enrich their paper.

1. In introduction, I suggest to the authors to explain why they choose this novel cyanobacteria to make the optimization of C-PE extraction. Is this a rich-PE cyanobacteria? Is this described in literature? I recommend tou to explain, even briefly why you choose this specie.

2. Regarding results, you have great focus on statistics, and that is ok. However, in my opinion there is a lack of presenting some results regarding the first factorial design. I would recommend you to present a table with conditions (runs) performed and the results regarding C-PE concentrations and purity index. For me and I guess for the readers too, would be easier to visualize the discrepancy/differences in results, than looking for the statistics itself.

3. In any time you compare your the C-PE concentrations obained in optimized conditions of extraction with literature. Is high, low? Have potential to be apply in any biotechnological field? To enrich your discusion, I sugest you to compare the concentrations obtained  with values  in literature...I felt you never refer to this...

Author Response

(The authors gave the same response as above.)
